# Nuts and Metabolic Syndrome: Reducing the Burden of Metabolic Syndrome in Menopause

**DOI:** 10.3390/nu14081677

**Published:** 2022-04-18

**Authors:** Celia Bauset, Ana Martínez-Aspas, Sara Smith-Ballester, Alicia García-Vigara, Aitana Monllor-Tormos, Fawzi Kadi, Andreas Nilsson, Antonio Cano

**Affiliations:** 1Service of Obstetrics and Gynecology, Hospital Clínico-INCLIVA, 46010 Valencia, Spain; cbausetc@gmail.com (C.B.); amaras82@hotmail.com (A.M.-A.); sarasmithballester@gmail.com (S.S.-B.); aliciagarcia@incliva.es (A.G.-V.); monllortormosaitana@gmail.com (A.M.-T.); 2School of Health Sciences, Örebro University, 701 82 Örebro, Sweden; fawzi.kadi@oru.se (F.K.); andreas.nilsson@oru.se (A.N.); 3Department of Pediatrics, Obstetrics and Gynecology, University of Valencia, 46010 Valencia, Spain

**Keywords:** menopause, metabolic syndrome, nuts, women

## Abstract

Menopause imposes a dramatic fall in estrogens, which is followed by an increase in the proportion of fat. The rising androgen/estrogen ratio along the menopause transition favors the accumulation of central fat, which contributes to insulin resistance and a series of concatenated effects, leading to a higher incidence of metabolic syndrome. The modulatory effect of diet on the metabolic syndrome phenotype has been shown for the Mediterranean diet, and nuts are key determinants of these health benefits. This review of the impact of nuts on the risk factors of the metabolic syndrome cluster examined studies—prioritizing meta-analyses and systemic reviews—to summarize the potential benefits of nut ingestion on the risk of metabolic syndrome associated with menopause. Nuts have a general composition profile that includes macronutrients, with a high proportion of unsaturated fat, bioactive compounds, and fiber. The mechanisms set in motion by nuts have shown different levels of efficacy against the disturbances associated with metabolic syndrome, but a beneficial impact on lipids and carbohydrate metabolism, and a potential, but minimal reduction in blood pressure and fat accumulation have been found.

## 1. Introduction

A joint statement by several scientific societies has defined metabolic syndrome (MetS) as a cluster of risk factors: (i) dysglycemia, (ii) increased blood pressure, (iii) dyslipidemia as defined by hypertriglyceridemia and low high-density lipoprotein (HDL), and (iv) central obesity [1]. MetS has gained in prominence due to its high prevalence worldwide and its association with risk of overt diabetes and cardiovascular disease (CVD) [2].

Central accumulation of fat, particularly visceral fat, plays a key role in inducing insulin resistance, which is a determining factor in metabolic syndrome (MetS) development [2,3]. Increased insulin resistance triggers dysglycemia and hyperinsulinemia, which also favors hypertension and dyslipidemia. Increased sodium reabsorption and retention, caused mostly by an activated renin–angiotensin–aldosterone system, is only one among various determinants of increased blood pressure by hyperinsulinemia [4]. Hyperinsulinemia also combines with alterations in the lipid profile associated with a drop in estrogens across menopause, to favor dyslipidemia [5,6,7].

Multiple clinical studies have addressed the question of whether menopause is associated with increased fat accumulation, loss of lean mass, or changes in body structure predominated by a trend to central fat increase. Ascertaining the specific role of menopause has created difficulty because menopause occurs at midlife, in a period during which several of these changes are induced by age per se in both men and women.

Using imaging technology such as computed tomography (CT) to measure visceral and subcutaneous abdominal fat, and dual-energy X-ray absorptiometry (DXA) to measure the proportions of lean and fat mass, researchers have confirmed that menopause triggers expansion of (total body and visceral) fat and a reduction in energy expenditure [8].

These findings have been confirmed in population studies. The Study of Women Across the Nation (SWAN) followed 1246 US women longitudinally from 8 years before until 10.5 years after menopause. Using DXA, investigators observed that women already experienced increases in both fat and lean mass in the years immediately preceding the menopausal transition. Subsequently, fat gain doubled but lean mass decreased, and this tendency was maintained until approximately two years post-menopause. An increase in bodyweight (BW) was also found, initiating prior to menopause and gaining speed across the menopausal transition [9].

Evidence of metabolic alterations has also been provided by a meta-analysis including both cross-sectional and longitudinal studies. Gains were observed in BW, body mass index (BMI), and percentage of body fat (BF). There was an increase in waist-to-hip ratio, visceral fat, and trunk fat percentage as well as an augmented abdomen/leg fat mass ratio [10]. Another study derived from the SWAN cohort confirmed that, consistent with the data on weight change, the risk for suffering MetS doubles in the years around menopause [11]. Diabetes risk is also increased, although there is some controversy about the responsibility of menopause for this [12].

Together with physical activity, healthy nutrition is the most recommended intervention for reducing the odds of developing MetS [13]. Evidence of the role of diet in modifying adverse cardiometabolic profiles during the menopausal transition was found in a longitudinal follow-up of 1246 midlife women (average age at baseline, 46.3 years) in the SWAN cohort. The Western pattern diet was associated with higher risk of carotid atherosclerosis, as defined by carotid intima-media thickness measurement [14]. Consistent with this premise, interest has recently grown in the role of diet in controlling the metabolic disturbances associated with menopause [15,16].

The positive impact of a healthy diet is a key message of the EAT-Lancet Commission, which has even proposed a universal reference healthy diet, with high consumption of fruit and vegetables and limited intake of meat and refined sugar among the main recommendations [17]. In this respect, the Mediterranean diet (MedDiet) has been proposed as one excellent alternative [18], which meets the health needs of women across the menopausal transition and after menopause [19]. MedDiet is not a strictly defined diet, but a selection of recommendations reflected in the food pyramid [20] (Figure 1).

Among the foods strongly recommended in the MedDiet, nuts and olive oil are receiving particular attention. Nut consumption is associated with a 15% reduction in the incidence of CVD and a 23% reduction in cardiovascular mortality [21]. Despite the interest derived from those data, there is limited information on the specific effect of nuts in crucial issues related to MetS, or its risk factors, across the menopausal transition and early post-menopause.

The aim of this review was to collect the available evidence on the modulatory effect that nut consumption may have on the risk of MetS, which increases across menopause. To improve the understanding of the process, we will first update the endocrinological substrate of metabolic changes across menopause. Given the limited availability of studies that directly investigate the effect of nuts on the MetS of menopause we used information obtained in the whole population.

**Figure 1 nutrients-14-01677-f001:**
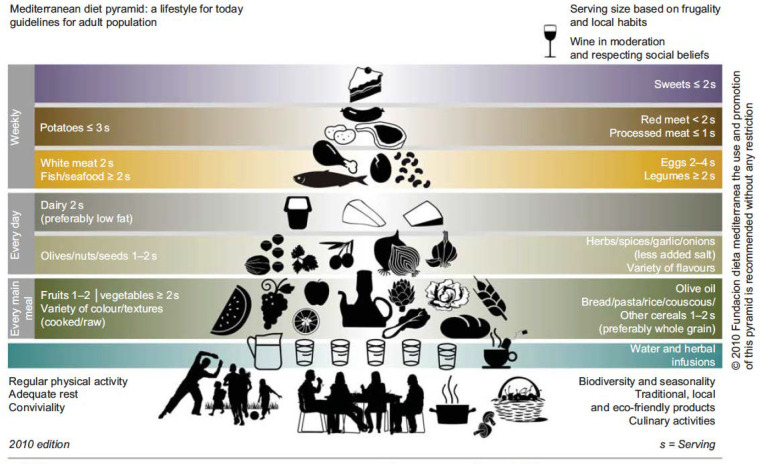
Principal components of the Mediterranean diet as represented by the diet pyramid. Intake frequency is shown, represented for each main food group. One or two servings of nuts are recommended daily. Reproduced by permission of Dr. L. Serra-Majem [22].

## 2. Menopause and Metabolic Disruption

The specific impact of menopause on increased BW and fat has raised questions about whether hormonal changes may be partly responsible. Menopause results from the depletion of ovarian follicles, a process stemming from continuous diminution of the follicle reserves acquired during fetal life [23,24]. The reduction of follicles to low numbers at menopause almost completely halts the production of estrogens [25]. The widespread distribution of estrogen receptors (ER) along different systems in the body underlies the association of menopause with other pathologies. Some key systems as such the cardiovascular apparatus and the central nervous system are important estrogen targets. Furthermore, age at menopause has been taken as a health biomarker according to epidemiological data showing that each year of later menopause is associated with a 2% decrease in all-cause mortality [26].

Androgens decline slowly with age, in a process that begins several years prior to menopause at both the ovary and the adrenal level [27]. Unlike estrogens, therefore, androgens do not undergo major changes across menopause, and this shifts the balance between estrogen and androgen levels in favor of the latter, thus subjecting women to a more androgenic milieu.

### 2.1. The Metabolic Impact of Estrogens and Androgens

#### 2.1.1. Estrogens

The specific role of estrogens in metabolic changes at menopause has been investigated at both the experimental and clinical level, with extensive use of genetically modified rodent models. Both inactivation of the aromatase gene to totally suppress estrogen synthesis, and inactivation of either ER type, α or β (ER knock-out, ERKO), have revealed an association between loss of estrogen action and fat accumulation. Increased number and size of adipocytes have been described [28]; changes have also been observed in bioenergetics, which has become more evident in models of surgical menopause, and additional findings include increased food intake and reduced spontaneous physical activity [28].

The conclusions obtained in animal models have been reproduced, although not unanimously, in clinical studies. Inducing transitory menopause with analogues of the gonadotrophin releasing hormone (aGnRH) has shown that reduction in energy expenditure related to a fall in estrogens was reverted by supplementation with estradiol (E2) [29]. Moreover, E2 alleviated the loss of fat-free mass and increase in central fat, including both subcutaneous and visceral compartments [30].

Some large clinical studies of menopausal hormone therapy (MHT) have included parameters related to BW and body composition indicators, although in no case as a main endpoint. The ad hoc methodology was therefore less robust and possibly for this reason, the evidence was more mixed. MHT showed beneficial changes by limiting BW gain and increases in waist circumference (WC) in the Postmenopausal Estrogen/Progestin Intervention (PEPI) trial [31] and in the Women’s Health, Osteoporosis, Progestin, Estrogen (HOPE) study [32]. Similarly, the Danish Osteoporosis Prevention Study also found that women randomized to MHT showed more limited weight increase than those receiving placebo. Interestingly, the difference was due to lower fat accumulation [33]. However, this finding is not consistent, and no benefit was observed in the Women’s Health Initiative (WHI) study [34]. Again, the subtle changes associated with hormonal action might have passed undetected in big studies with other principal endpoints.

#### 2.1.2. Androgens

Androgens have also been shown to have a metabolic impact, which follows a paradoxical direction between the two sexes. A study on participants in the UK Biobank study showed that genetically higher testosterone was beneficial in men, where it reduced the risk of type 2 diabetes (T2D), but was detrimental in women, increasing the risk of T2D and polycystic ovary syndrome (PCOS) [35]. In a cross-sectional study on women with PCOS other investigators found that free, but not total, testosterone was related to WC and insulin resistance [36].

The more androgenic milieu of menopause may thus favor fat accumulation, with a trend towards central fat. This may add to the abovementioned impact of lower estrogen levels and contribute to the tendency towards fat accumulation and the central obesity observed at menopause. The size of this effect is however not yet fully understood.

## 3. Nut Components of Health Interest

Nuts is an umbrella term covering variants such as walnuts, hazelnuts, almonds, peanuts, pistachios, cashews, and pine nuts, with most published research focused on walnuts.

Nuts are high calorie foods, but despite this, the properties of their components have been linked to beneficial health effects in both experimental and clinical studies. Obviously, these benefits vary somewhat from one type to another, as mediated by differences in composition between nut types, but a general composition profile can be identified [37]. This includes (i) macronutrients, with a relatively high proportion of unsaturated fat, monounsaturated fatty acids (MUFAs) and polyunsaturated fatty acids (PUFAs) compared to carbohydrates or protein; (ii) bioactive components, including phenolic compounds, as well as phytosterols, antioxidant vitamins and minerals, and (iii) fiber.

### 3.1. Unsaturated Fat

This abundance of unsaturated fat is shared with other fats highly represented in the MedDiet, such as olive oil, although nuts have certain specificities like higher proportion of PUFAs. Specifically, the main fatty acid in nuts is linoleic acid, a PUFA accounting for 40–60% of total fatty acid composition in several types of nuts. The other two main fatty acids are oleic acid and α-linolenic acid [38], the latter being particularly concentrated in walnuts. Both MUFA and PUFA confer advantages over saturated fat in several endpoints, such as lipid profiles, inflammation, and of specific interest for metabolic syndrome and CVD, oxidation [39]. Their anti-inflammatory properties have been confirmed in the Multi-Ethnic Study of Atherosclerosis (MESA), a longitudinal clinical study including 6000 participants from the general population. Consumption of nuts was inversely associated with circulating levels of inflammatory markers such as C-reactive protein (CRP), interleukin 6 (IL-6), fibrinogen, etc. [40]. These findings have been also reproduced in studies including individuals at high cardiovascular risk [41]. Translating these mechanisms into real primary and secondary protection of CVD, however, is yet to be attained, as concluded in a Cochrane review of randomized controlled trials (RCT). A slight reduction in cardiovascular events, possibly mediated through lipid reduction, may occur, but more evidence is required [42].

### 3.2. Polyphenols

Like olive oil, nuts are also rich in polyphenols, a large group of bioactive phytochemicals characterized by a molecular structure containing phenol rings. This family includes a long list of sub-classes, the most significant of which are flavonoids, phenolic acids, stilbenes, and lignans [43,44]. Due to the abundant number of species included within the umbrella term nuts, the list of polyphenols in nuts is also long, including flaxseed as a main component in lignans, hazelnuts, and pecans. Almonds are rich in proanthocyanidins, and chestnuts and walnuts are rich in ellagitannins [45].

One significant feature of polyphenols is their metabolism in the intestine by local microbiota. This process dramatically amplifies the number of metabolites, which range from 100,000 to 200,000. The metabolic process and subsequent passage to plasma of the resulting products are quick [46].

The lack of an established direct relationship between inadequate intake of plant polyphenols and specific diseases, in contrast to what is known of vitamins and other nutrients, precludes as yet reference sources of these food compounds. Evidence, nonetheless, exists of a health benefit for polyphenols found in different fruits, cocoa, coffee, etc. [47], and action on some disease mechanisms like oxidative stress and inflammation has also been reported for these phytochemicals [48,49].

### 3.3. Other Components

Nuts are also a source of minerals, vitamins, and other salutary nutrients. Magnesium, calcium, potassium, and phosphorus comprise the main types of minerals, while vitamin E or B3 make up the main vitamins. Ammonium compounds like choline may be also found in nuts. The main components of different types of nuts are shown in Table 1.

## 4. Literature Search

An electronic search was performed in the PubMed database for articles published from 1 January 2000 up to 15 December 2021. The relevant terms (nuts OR walnut OR almond OR peanut OR pistachio OR cashew OR hazelnut OR pine nut) were paired with the following keywords (metabolic syndrome OR hypertension OR blood pressure OR diabetes OR glycemia OR carbohydrates OR lipids OR cholesterol OR HDL OR LDL OR triglycerides). We further restricted the search using language filters to retrieve only articles published in English or Spanish and filtering for species to include only humans.

Conclusions from meta-analyses and systematic reviews (SR) were prioritized when available. Original studies were included when published subsequent to the most recent meta-analysis or when conferring particular impact because of size, length of follow-up, or other features of importance.

The search yielded 3798 references. Title, or title and abstract when the title raised uncertainty about the content of the paper were examined by five investigators (CB, AM-A, SS-B, AG-V, and AM-T). Disagreements between them were settled by consensus with the corresponding (AC) and the two other senior authors (FK and AN). The selection included a total of 263 papers, together with 44 articles obtained by hand-search of reference lists of the selected papers. Finally, 307 papers were reviewed in depth, and, from these, 120 papers were selected for citation. The flowchart of the literature search is represented in Figure 2.

## 5. Effect of Nuts and Their Components on Metabolic Syndrome

Data on the effects of nuts on MetS has focused both on mechanisms of action set in motion by either nuts or their constituent compounds, and on clinical impact. While the former frequently use experimental models, evidence in the latter stems from observational or interventional studies. Current literature is concentrated on MetS itself, or more frequently on some of its components.

Most data obtained stems from studies using nuts in general, without distinction between variants, but there is also a growing number of studies on the role of specific nut types, mainly walnuts or peanuts, which in some cases have been used as extracts.

Experimental studies have revealed a variety of potential mechanisms of action set in motion by specific compounds in nuts [50]. Clinical evidence has become progressively stronger in recent years thanks to the availability of meta-analyses. Even so, consistent conclusions are often limited by the low number of studies, most of them small and with conflicting results.

### 5.1. Mechanisms Affecting Components of MetS

#### 5.1.1. Body Weight and Waist Circumference

An increase in both BW and WC can be expected from intake of high fat foods such as nuts. Nonetheless, some mechanisms may oppose that effect, specifically (i) the satiety effect of fiber, which may lead to a reduction in food intake, (ii) data suggesting that unsaturated fat and protein in nuts are associated with a higher metabolic rate, thus increasing energy expenditure [51,52], (iii) the contribution of gut microbiota through a possible prebiotic effect of nuts [53], and (iv) the potential contribution of polyphenols, which may theoretically act through several mechanisms such as an increase in lipolysis or reduction in adipocyte differentiation, although their impact in practice is uncertain [54].

#### 5.1.2. Carbohydrate Metabolism

Mechanisms with a protective action in the metabolism of carbohydrates include the previously mentioned fiber content, which favors a reduction in food intake by generating satiety. This is in addition to the possible prebiotic effect of polyphenols and also fiber, with a presumed impact on microbiota and favorable consequences on glucose metabolism [50]. Nuts also have a low glycemic index owing to a higher concentration in fat, leading to a lower contribution of glucose to the energy supply; therefore, less insulin is required, and this should favor control of diabetes [37].

One important benefit particularly associated with flavanols, a sub-type of flavonoid, is the reduced risk of prediabetes and T2D. Different mechanisms have been proposed, including a reduction in insulin resistance which translates into increased uptake of glucose by the target tissues [45]. In the particular case of nuts, ellagic acid, present in several nuts in the form of ellagitannins, has been attributed to a protective effect against diabetes [55].

Additional mechanisms have been pinpointed from experimental models (reviewed in [41]). For example, α-linolenic acid increases secretion of the glucagon-like peptide-1—an incretin that decreases blood glucose levels—by pancreatic β-cells in rodents [56]. Several studies have focused on actions linked with gene–diet interactions, including transcriptomic and epigenomic mechanisms [57]. For instance, a normocaloric diet enriched with PUFA from almonds and walnuts demonstrated an impact in the profile of 192 common micro-ribonucleic acids (miRNAs) [58]. In another study, long-standing pistachio consumption positively modulated a set of miRNAs related to glucose metabolism and insulin resistance [59].

#### 5.1.3. Blood Pressure

The limitation on the aforementioned increased sodium reabsorption and retention caused by insulin is reduced as a result of increased insulin sensitivity [4]. As an additional mechanism, magnesium, a chemical component abundant in nuts, may stimulate production of vasodilatory mediators such as prostacyclin in addition to other mechanisms [60,61]. Finally, the already mentioned reduction of BW also contributes to a decrease in blood pressure.

#### 5.1.4. Lipids

Four important components of nuts (polyphenols, unsaturated fat, phytosterols, and fiber) have shown detectable effects on lipids [62]: (i) ellagitannin, a type of polyphenol found in nuts, has been associated with reduction of low-density lipoprotein (LDL) and triglycerides (TG) and small increases in HDL, but the changes seem to concentrate in obese/overweight people [63]; (ii) the effect of unsaturated fat is supported by a cause–effect link between MUFA and PUFA components in walnuts and maintenance of LDL levels, which has been endorsed by the European Food Safety Authority [64]); (iii) phytosterols, a family of plant compounds with a similar molecular structure to cholesterol, have shown hypolipidemic action [65]; (iv) finally, fiber decreases the process of gastric emptying, impairs diffusion in the small intestine, and increases bile acid excretion.

### 5.2. Clinical Studies

#### 5.2.1. Metabolic Syndrome

Two relevant studies have taken MetS as an endpoint. A sub-group of participants (n = 5800) in the Prevención con Dieta Mediterránea (PREDIMED)-Plus study was analyzed after one year for changes in risk factors for the MetS. Nut consumption preceded an improvement in MetS, as derived from a reduction in WC, BW, BMI, and systolic blood pressure (SBP), and an increase in HDL [66]. The Finnish Diabetic Nephropathy study, in turn, investigated nut consumption in participants diagnosed with type 1 diabetes, and found that the 369 participants having ≥2 servings of nuts experienced favorable changes in all the components of MetS [67].

#### 5.2.2. Body Weight and Waist Circumference

##### Nuts in General

A large epidemiological study comprising three separate cohorts including 120,877 American participants concluded that regular consumption of nuts was associated with a slight reduction of 0.26 kg at 4-year follow-up [68]. This apparently paradoxical effect, given the high caloric composition of nuts, has stimulated an increasing number of studies providing evidence that has subsequently been assessed in recent meta-analyses. A growing proportion of published literature consists of RCT, but again, studies are small in size, which may explain the conflicting results.

A dose–response meta-analysis of prospective observational studies until August 2018 found an inverse association between nut consumption and abdominal obesity (summary relative risk (RR) = 0.42, 95% confidence intervals (CI): 0.31, 0.57), but no association was found for overweight risk or weight gain. The authors warned that the quality of the evidence was low or very low [69]. Four subsequent meta-analyses have assessed RCT on the effect of nuts to conclude that no effect was detected in BW, BMI, percentage of BF, or WC [70,71,72,73], and a reduction in hunger was found but without effect on fullness [70]. Another meta-analysis including both prospective studies (n = 569,910) and RCT (n = 5873) concluded, however, that higher nut intake was related with a reduction in BW and BF [74].

##### Specific Nut Types

The particular effect of walnuts was addressed in another meta-analysis, which also gave neutral results, although a dose–response effect could be observed through intake up to 35 g/day [75]. Similarly, no significant impact was confirmed for pistachios [76], with the exception of an isolated reduction in BMI [77], and the same neutral results were found for cashews [78]. The specific impact of almonds has been evaluated in a meta-analysis of RCT which detected a summary net change of −1.39 kg (95% CI: −2.49, −0.30 kg) in BW [79].

##### Later Clinical Evidence

We also retrieved several subsequently published studies not included in the aforementioned meta-analyses. Nuts and pistachios were analyzed in RCT, with no effect detected apart from the sensation of satiety, which would presumably favor adherence in weight reduction programs [80,81,82].

Although observational, the size (n = 373,293 subjects from 10 European countries) and 5-year follow-up period raised the interest in the European Prospective Investigation into Cancer and Nutrition—Physical Activity, Nutrition, Alcohol, Cessation of smoking, Eating out of home in relation to Anthropometry (EPIC-PANACEA). While all participants gained an average of 2.1 kg, those in the highest quartile of nut intake gained less weight (overall mean −0.07 kg; 95% CI: −0.12, −0.02 kg) than non-consumers [83].

The PREDIMED study is of special interest as an RCT intervention trial, in which participants (n = 7447) were asymptomatic older people (men aged 55–80 years and women 60–80 years) at risk of CVD. All the three arms experienced a slight reduction in BW and an increase in WC, but the group randomized to nuts experienced a lower WC increase than the control group (adjusted difference, −0.94 cm) [84].

In conclusion, it seems that nuts are not associated with the increase in BW expected from their high calorie content. Indeed, a slight reduction in weight gain may take place, which shows a more pronounced effect in waist increase. The evidence, however, is still of low quality.

#### 5.2.3. Carbohydrate Metabolism and Diabetes

##### Nuts in General

Acute feeding trials have shown that nuts decrease postprandial hyperglycemia in a dose-dependent manner in both healthy and diabetic subjects [85]. Major epidemiological studies such as the Nurses’ Health Study and the Shanghai Women’s Health Study have reported a reduction of up to a third in T2D diagnoses, but, in contrast, no effect was observed in the Iowa Women’s Health Study or the Physician’s Health Study (reviewed in [85]).

A systematic review and meta-analysis examined databases from 25 observational studies and 2 RCTs; the authors concluded that eating nuts was inversely associated with diagnosis of diabetes (RR = 0.87, 95% CI: 0.81, 0.94) [86]. However, a review of meta-analyses of both observational and RCT could not reproduce an impact on diabetes diagnosis, although a reduction in fasting blood glucose (FBG) was found in the three meta-analyses of intervention studies [87]. This same conclusion was found in a more recent meta-analysis of observational studies [88].

Recent years have witnessed an increase in RCT in this research area. One meta-analysis found intake of tree nuts and peanuts to be associated with a significant reduction in the homeostatic model assessment (HOMA-IR) and fasting insulin (FI) (weighted mean difference, WMD = −0.40 μIU/mL, 95% CI: −0.73, −0.07 μIU/mL), but not in FBG [89], an observation that was confirmed in two other meta-analyses limited to adults with pre-diabetes, this time including as a finding a decrease in FBG [90] and also T2D, although the effect was small [91].

##### Specific Nut Types

Studies have been published reporting meta-analyses on the specific action of certain types of nuts. Almonds were found to reduce FBG when consumption exceeded 42.5 g/day [79]. The effect on individuals with T2D was mixed, since a reduction was found in glycosylated hemoglobin (HbA1c), but not in FG or FI or insulin resistance, as assessed by the HOMA-IR [92]. One more recent RCT not included in the meta-analyses found reduced FG with intake of almonds compared with sweet biscuits for snacks [93].

Walnut consumers were concluded to have lower risk of diabetes than non-nut consumers (odds ratio, OR = 0.47; 95% CI: 0.31, 0.72) in an assessment of the 24-h diet in adults included in the National Health and Nutrition Examination Survey (NHANES) database [94]. More recent meta-analyses on RCT include an assessment of small studies with specificities regarding selected population or type of intervention. With small differences, they reported a neutral effect on FG, FI, HbA1c, or HOMA-IR for walnuts [95,96], cashews [78], peanuts [97], and pistachios [98].

##### Later Clinical Evidence

A more recent RCT derived from a sub-analysis of the PREDIMED study (n = 1833) found a reduced incidence of T2D associated with walnut consumption [99].

In conclusion, various clinical studies seem to point to a benefit of nuts in carbohydrate metabolism. This is further supported by the majority of meta-analytical data analyses, although there are some conflicting results and the size of the effect, when present, seems small.

#### 5.2.4. Blood Pressure

##### Nuts in General

Findings from epidemiological studies taking blood pressure as a primary objective are not unanimous [100,101], and interventional studies also show heterogeneity with regard to type and dose of nut or population profiles (reviewed in [85]). Reflecting that uncertainty, a meta-analysis of 17 RCT concluded that dietary patterns that include nuts in their composition such as MedDiet, the Dietary Approaches to Stop Hypertension (DASH) diet, and the Nordic diet, significantly decreased SBP and diastolic blood pressure (DBP) [102], but another systematic review and meta-analysis of 61 controlled trials could not detect any effect [103]. An inconsistent impact on blood pressure was also found in a systematic review and network meta-analysis of 66 randomized trials identifying the role of distinct food groups in blood pressure [104] as well as in a review of meta-analyses of prospective and interventional studies [87].

##### Specific Nut Types

More recently, several meta-analyses on RCT concerning the effect of specific types of nuts have been published. A systematic review and meta-analysis that restricted the scope to RCT on almond consumption also confirmed a neutral impact on blood pressure [79]. A small reduction has been found for SBP with pistachios [76,98] and cashews [105], while DBP reduction was found for almonds [106] and no effect for walnuts [107].

##### Later Clinical Evidence

A salient intervention trial because of its design and size, the aforementioned sub-study from PREDIMED included 5800 overweight men and women with MetS followed for one year. Investigators confirmed that nut consumption was related to a reduction in SBP only [66].

In summary, while dietary interventions in which nuts are a crucial component—such as PREDIMED—seem to be associated with reduction in blood pressure in any form, the overall assessment of studies dealing with nuts as the single factor seems to show a mostly neutral effect.

#### 5.2.5. Lipids

##### Nuts in General

Nuts have been shown to decrease total cholesterol (TC), LDL, and TG in individuals free of cardiovascular risk, as described in two systematic reviews and meta-analyses on controlled intervention trials [103,104]. The reduction in TC and LDL has been also confirmed by a review of meta-analyses [87]. The impact on TG has been limited to hyper-triglyceridemic subjects in other analyses [108].

##### Specific Nut Types

Several meta-analyses aiming at disclosing the particular effects of specific types of nuts in RCT have been published, but the evidence is limited due to low participant numbers in the published analyses. Pistachios and walnuts [109,110] have been considered more active in favoring a beneficial lipid profile, due to actions on TC, LDL, and TG. Additional evidence on pistachios has confirmed a reduction in TC, LDL, and TG, with no change in HDL [111], the TC/HDL ratio or the LDL/HDL ratio [98]. Almonds reduced TC and LDL, the effect being small, from 5.92 mg/dL to 4.92 mg/dL (TC) and from 4.83 mg/dL to 5.65 mg/dL (LDL) [112,113]. There was a concomitant decrease [112,113] or no change in TG [76] and no change [112,113] or a small reduction in HDL [79]. The same effect was found for peanuts, but a parallel increase was detected for both TC and HDL [97]. Cashews, however, showed no effect on lipids [105,114].

##### Later Clinical Evidence

Some more recent RCT have been published in the last few years, some of which are relatively small trials investigating the effect of pecan nuts, almonds, walnuts, or nuts in general. Overall, benefits in lipid pattern were similar or with small changes to reported findings in the recent meta-analyses. Specifically, the effect of 30 g/day pecan nuts was studied in 204 individuals with stable coronary artery disease for 12 weeks. The 68 participants having pecan nuts exhibited a significant reduction in non-HDL cholesterol levels (12.1 mg/dL) vs. the control and in the TC/HDL ratio (0.3 vs. the control) [115]. The effect of almonds was investigated in the Almonds Trial Targeting Dietary Intervention with Snacks (ATTIS) study, in which 107 adults at above-average risk of CVD were randomized to whole roasted almonds (n = 51) or control snacks (n = 56), providing 20% of daily estimated energy requirements for 6 weeks. The only lipid benefit consisted of a reduction of LDL (9.75 mg/dL in the almond group) [116]. The lipoprotein response to a low-calorie (25% of energy deficit) diet was investigated in 67 stable coronary artery disease individuals who were randomized to a nut-containing or nut-free arm. After 8 weeks follow up, the intake of nuts was linked with a significant increase in HDL (1.17 mg/dL) and apolipoprotein A1 (2.55 mg/dL) [117].

Two larger RCT have been added. The PREDIMED-Plus study, which followed 5800 men and women for one year, found that increased nut consumption was followed by a reduction in TG (7.8 mg/dL) and an increase in HDL (1 mg/dL in women vs. 1.9 mg/dL in men) [66]. The Walnuts and Healthy Aging (WAHA) study, which followed 628 subjects for 2 years, found that the consumption of walnuts was related to a reduction in TC and LDL, while no change was detected for HDL and TG. As for PREDIMED, there was a sex-specific effect in LDL reduction, in which, for unknown reasons, women experienced a −2.6% change vs. −7.9% in men [118].

Additional PREDIMED data have shown that nuts may improve HDL-dependent reverse cholesterol transport [cholesterol efflux capacity] [119]. This parameter offers information about the functional capacity of HDL, a quantity that reflects the anti-atherogenic potential of HDL better than HDL levels. Specifically, the cholesterol efflux capacity mainly reflects how efficient HDL is in performing the extraction of excess cholesterol from peripheral cells to the liver. Finally, several studies have included changes in apolipoproteins, including reduction in apolipoprotein B [103] independent of nut type or associated diet.

To conclude, most of the available evidence favors a protective effect on the lipid profile. This effect of nuts on lipids has been endorsed by the American Heart Association (AHA), which includes nut consumption in dietary strategies for controlling cholesterol [120].

## 6. Conclusions

The increased incidence of metabolic syndrome in women associated with menopause can be mitigated by a healthy lifestyle, including a healthy diet. Nuts are a major component of the Mediterranean diet, which has been selected as a first-choice option for promoting diet-related intervention.

There are several species variants in the nut family, but all maintain quite a stable nutrient composition: specifically, unsaturated fat, bioactive (particularly phenolic-related) compounds, and fiber. The benefits associated with each nut constituent relate to general disease mechanisms such as oxidative stress or inflammation, and changes in microbiota. A wealth of experimental studies has described likely protective effects against disturbances in carbohydrate metabolism, lipid profile, blood pressure, and fat accumulation. Confirmation in clinical studies is still a process under development, with often insufficient evidence due to the limited quality of available studies. A beneficial impact on lipids and on carbohydrate metabolism seems a more solid conclusion, and other plausible effects are a potential but minimal reduction effect on blood pressure, together with a likely reduction in the process of fat accumulation and increase in waist circumference associated with menopause.

## Figures and Tables

**Figure 2 nutrients-14-01677-f002:**
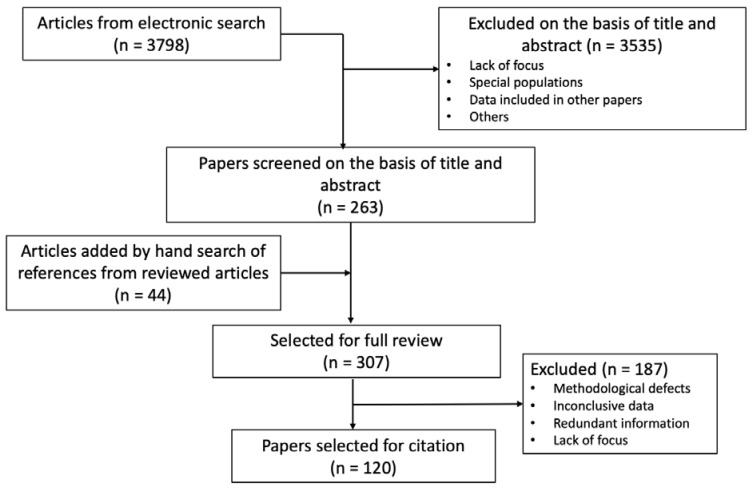
Flowchart of the literature search.

**Table 1 nutrients-14-01677-t001:** Main nutrients in different types of nuts (per 100 g of raw nut) [37].

Nutrient	Almonds	Cashews	Hazelnuts	Peanuts	Pecans	Pistachios	Walnuts
Total fat (g)	49.9	43.9	60.8	49.2	72.0	45.3	65.2
SFA (g)	3.8	7.8	4.5	6.3	6.2	5.9	6.1
MUFA (g)	31.6	23.8	45.7	24.4	40.8	23.3	9.0
PUFA (g)	12.3	7.8	7.9	15.6	21.6	14.4	47.2
CHO (g)	21.6	30.2	16.7	16.1	13.9	27.2	13.7
Protein (g)	21.2	18.2	15.0	25.8	9.2	20.2	15.2
Fiber (g)	12.5	3.3	9.7	8.5	9.6	10.6	6.7
Ca (mg)	269	37	114	92	70	105	98
K (mg)	733	660	680	705	410	1025	441
Mg (mg)	270	292	163	168	121	121	158
P (mg)	481	593	290	376	277	490	346
Phytosterols (mg)	197	151	122	NA	158.8	214	110.2
Total phenols (mg)	287	137	687	406	1284	867	1576
Vitamin E (mg)	25.6	0.9	15.0	8.3	1.4	2.9	0.7
Energy (Kcal)	579	553	628	567	691	560	654

Original information obtained from the United States Department of Agriculture (USDA). CHO, carbohydrates; K, potassium; Mg, magnesium; MUFA, monounsaturated fatty acids; NA: not available; P, phosphorus; PUFA, polyunsaturated fatty acids; SFA, saturated fatty acids.

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
