# Peer review of "Nuts and Metabolic Syndrome: Reducing the Burden of Metabolic Syndrome in Menopause"

_nutrients, 2022, doi:10.3390/nu14081677_

Round 1
Reviewer 1 Report
Dear authors, the manuscript is very well structured and written. However, some comments should be addressed before their publication in Nutrients.
- L176. What was the criteria for select the relevant terms. Please specify
- L463. Please add more information about how HDL-dependent reserve cholesterol trasnport-cholesterol efflux capacity.
- Please double check the References section.
- Reference 2, 4, 9, 12, 13, 14, 18, 22, 25, 32, 34, 33, 40, 48, 52, 53, 55-58, 64, 66, 69, 70, 72, 74, 76, 84, 85, 90, 95, 96, 102, 105, 116, 120, 123, 124. The format is different to the rest. Please correct the mistakes.
Author Response
Reviewer #1.-
Many thanks for your comments, which have improved the quality of our manuscript.
Dear authors, the manuscript is very well structured and written. However, some comments should be addressed before their publication in Nutrients.
Many thanks for your comment. We appreciate it.
- L176. What was the criteria for select the relevant terms. Please specify
We firstly considered and debated which was the best approach to the issue. We found that there is plenty of literature about the metabolic impact of nuts in areas related with the metabolic syndrome. Furthermore, it is obvious that menopause associates with important metabolic changes favoring the increase in the prevalence of metabolic syndrome. There is , however, a lack of papers reporting results of research about the impact of nuts on the metabolic syndrome of menopause. Therefore, we decided to update the main features of the metabolic changes leading to increased prevalence of metabolic syndrome across menopause, and then those of nuts in what refers the impact on the metabolic syndrome.
So, the search was designed in such a way that a list of the main types of nuts, which was also added the generic term of nuts, was crossed with the available literature about the metabolic syndrome. In order to pick up each relevant publication we included the 4 components, hypertension (hypertension or blood pressure), dysglycemia (diabetes or glycemia or carbohydrates), dyslipidemia (lipids, or cholesterol or HDL or LDL, or triglycerides) together with the generic “metabolic syndrome”.
We did not include the term “menopause” because the list of papers was dramatically reduced given the limited number of publications addressing the issue. Also, such an approach would miss the analysis of the main determinants of nuts having an impact, and which that impact is, on the metabolic syndrome.
- L463. Please add more information about how HDL-dependent reserve cholesterol trasnport-cholesterol efflux capacity.
We agree with the reviewer that it is helpful for readers that a detailed explanation of what the efflux capacity is added. So, we have included two new sentences, as follows:
“This parameter of HDL offers information about the functional capacity of HDL, a magnitude that reflects the anti-atherogenic potential of HDL better than HDL levels. Specifically, the cholesterol efflux capacity mainly reflects how efficient is HDL in performing the extraction of excess cholesterol from peripheral cells to the liver”.
- Please double check the References section.
- Reference 2, 4, 9, 12, 13, 14, 18, 22, 25, 32, 34, 33, 40, 48, 52, 53, 55-58, 64, 66, 69, 70, 72, 74, 76, 84, 85, 90, 95, 96, 102, 105, 116, 120, 123, 124. The format is different to the rest. Please correct the mistakes.
We are sorry that have not been able to identify where the loss of the format is. We find the format agrees with the authors rules of Nutrients and that follow the same format as the others.
Reviewer 2 Report
Dear Authors,
The reviewed article is a review on the impact of dietary supplementation with various variants of nuts on the metabolic syndrome. The discussed topic is interesting and worth exploring.
Nevertheless, according to the reviewer's opinion, the content of the article does not correspond to its title. The title Effect of nuts on the metabolic syndrome associated with menopause suggests that the reader will be dealing with results related to menopausal or near-menopausal women. Meanwhile, this review is based on articles on dietary supplementation for both men and women of different ages (age is not given exactly, which is also a gap in the data provided). The literature search description (Chapter 4) is based on different keywords (value words), but the basic word is missing - manopause !!! Therefore, the article should be redrafted in such a way that the content corresponds to the title or vice versa, which is left for the authors' decision. I propose changing the title of the work and introduction (omit the concept of menopause in theoretical considerations) and focusing on the metabolic syndrome that affects people in general.
Additionally, below are some suggestions that I hope will help improve this work.
- In the introduction, there is no purpose of the work
- There are many abbreviations used in the work - it is suggested to make a list of them
- Chapter 5 NUTRITION INGREDIENTS OF HEALTH INTEREST is not detailed enough. Only unsaturated fats and polyphenols are described, the others are only mentioned. Nuts are a source of vitamins (especially vitamin E, B vitamins) and minerals (P, Mg, Fe, Ca, K, Zn, Na ...). In this chapter, it is worth comparing the composition of individual types of nuts in the table. Additionally, I suggest moving this chapter before Chapter 4. LITERATURE SEARCH
- Chapter 4 should not be bold
- Figure 2. - arrow pointing away from the block Articles added by hand search of references from reviewed articles (n=44)
- Describing the influence of the consumption of nuts on various parameters, I propose to present the data in more detail. For example:
- line 435 - the authors write that it is not the type of nuts that matters, but the dose (what is the optimal dose?),
- line 444 almonds lower TC and LDL - how big is this difference compared to the control or pre-supplementation value,
- line 453 benefits in lipid pattern were found - how has this changed?
- In the description of individual clinical trials, there is no information on the studied cohort (sex, age), type and dose of supplemented nuts, method and time of their supplementation.
Author Response
Reviewer #2.-
Many thanks for tour suggestions, which have improved the quality of our manuscript.
*The reviewed article is a review on the impact of dietary supplementation with various variants of nuts on the metabolic syndrome. The discussed topic is interesting and worth exploring.
Thank you for the comment.
*Nevertheless, according to the reviewer’s opinion, the content of the article does not correspond to its title. The title Effect of nuts on the metabolic syndrome associated with menopause suggests that the reader will be dealing with results related to menopausal or near-menopausal women. Meanwhile, this review is based on articles on dietary supplementation for both men and women of different ages (age is not given exactly, which is also a gap in the data provided). The literature search description (Chapter 4) is based on different keywords (value words), but the basic word is missing - manopause !!! Therefore, the article should be redrafted in such a way that the content corresponds to the title or vice versa, which is left for the authors' decision. I propose changing the title of the work and introduction (omit the concept of menopause in theoretical considerations) and focusing on the metabolic syndrome that affects people in general.
Thank you for the remark. We now detail the answer to each point:
Menopause is missing as a keyword.-
We have found the difficulty that the published literature does not take menopause as a variable, so the best way we found to address the issue was to review the topic and then extend the body of knowledge to the metabolic syndrome of menopause.
This also affects the other concern of the reviewer about the selected keywords. Because of the lack of studies properly focused on menopause, the inclusion of that keyword leads to a very poor search outcome. Indeed, we checked it and only found 47 papers. After reviewing the titles, only 1 seemed appropriate, the paper by Fantino et al, which has been already included in our manuscript (see ref 79). More detail is given above, in the corresponding answer to reviewer#1.
Title and introduction need to be changed
We agree that the title needs to be changed, so the new one is: Nuts and the metabolic syndrome. Lessons to reduce the burden of the metabolic syndrome of menopause.
We have also omitted the conceptual description of menopause in the original Introduction. Now we have prepared a new one in which we directly go to the point of the metabolic syndrome definition, and then go to the link of metabolic syndrome and menopause; subsequently, we claim the interest of nutrition in that regard and have created a new paragraph (see lines 94-99) to present the purpose of the review, as suggested by the reviewer.
This is followed by a new section in which we present the endocrinological background supporting the metabolic changes found at menopause.
Additionally, below are some suggestions that I hope will help improve this work.
- In the introduction, there is no purpose of the work
Many thanks again. As mentioned above, we have introduced it now (lines 94-99)
- There are many abbreviations used in the work - it is suggested to make a list of them
This has been done as well.
- Chapter 5 NUTRITION INGREDIENTS OF HEALTH INTEREST is not detailed enough. Only unsaturated fats and polyphenols are described, the others are only mentioned. Nuts are a source of vitamins (especially vitamin E, B vitamins) and minerals (P, Mg, Fe, Ca, K, Zn, Na ...). In this chapter, it is worth comparing the composition of individual types of nuts in the table. Additionally, I suggest moving this chapter before Chapter 4. LITERATURE SEARCH
We appreciate the remark. We have now moved the section before chapter 4. Moreover, we have added a table (actual Table1), which presents a very detailed content list including the main nutrients in each of the main nuts types.
- Chapter 4 should not be bold
This has been amended as well.
- Figure 2. - arrow pointing away from the block Articles added by hand search of references from reviewed articles (n=44)
This has been amended now (see new Figure2)
- Describing the influence of the consumption of nuts on various parameters, I propose to present the data in more detail. For example:
- line 435 - the authors write that it is not the type of nuts that matters, but the dose (what is the optimal dose?),
Thanks for the comment. We agree with the reviewer that such a general remark makes little sense, particularly because the immediate question in the reader will be about the optimal dose, something that cannot be stated given the variety of nuts types and outcomes. So we have omitted the sentence.
- line 444 almonds lower TC and LDL - how big is this difference compared to the control or pre-supplementation value,
We agree with the reviewer in that, given the relevance of the lipid effects of nuts, those studies need to be detailed. We have done so for each study (see lines 444-456). We have omitted reference 120 concerning walnuts because this is a complex study which mainly focuses in the microbiota, and in which the lipids effects are secondary. So a detailed description would be confusing and out of the focus of the manuscript.
- line 453 benefits in lipid pattern were found - how has this changed?
This comment connects with the previous one. We have now described the main features of each of the mentioned studies.
-In the description of individual clinical trials, there is no information on the studied cohort (sex, age), type and dose of supplemented nuts, method and time of their supplementation.
As mentioned above, we have now added a description of the cited studies. This applies to the mentioned lines 444-456, but also to lines 457-464.
Round 2
Reviewer 2 Report
Deat Authors,
thank you for the comprehensive explanations and corrections made.
I just wanted to pay attention to the editorial changes, e.g. chapters 1-3 are written with justified text, with no gaps between paragraphs and with the indentation of the first line of the paragraph, while the text of the remaining chapters has been left-aligned and there are gaps between the paragraphs.
The numbering of references should also be changed.